



# CROPGRIDS: A global geo-referenced dataset of 173 crops circa 2020

Fiona H. M. Tang[1], Thu Ha Nguyen[2], Giulia Conchedda[3], Leon Casse[3], Francesco N. Tubiello[3], Federico Maggi[2,4]

[1]School of Environmental and Rural Science, University of New England, Armidale, New South Wales 2351, Australia.
[2]Environmental Engineering, School of Civil Engineering, The University of Sydney, Sydney, New South Wales, Australia.
[3]Statistics Division, Food and Agriculture Organization of the United Nations, Viale delle Terme di Caracalla, Rome 00153, Italy.
[4]Sydney Institute of Agriculture, The University of Sydney, Sydney, NSW, 2006, Australia

*Correspondence to*: Federico Maggi (federico.maggi@sydney.edu.au)

**Abstract.**

Despite recent advancements in cloud processing and modelling and the increasing availability of high spectral- and temporal-resolution satellite imagery, mapping the spatial distribution of crop types remains a challenging task. Here, we present CROPGRIDS – a comprehensive global, geo-referenced dataset providing information on areas for 173 crops circa the year 2020, at a resolution of 0.05˚ (~5.55 km at the equator). It represents a major update of the Monfreda et al. (2008) dataset, the most widely used geospatial dataset previously available, covering 175 crops with reference year 2000 at 10 km spatial resolution. CROPGRIDS updates Monfreda et al. (2008) through the careful evaluation of 26 published gridded datasets covering more recent crop information at regional, national, and global levels, largely over the period 2015 – 2020. The new product successfully updates the area extent for 80 of the 175 crops originally covered, representing an update to 1.2 billion hectares of crop area (i.e., 81% of the total cropland included in CROPGRIDS). CROPGRIDS carries forward the crop type maps originally in Monfreda et al. (2008) for 93 crops as more recent information for these crops is not available. We compared CROPGRIDS harvested area of individual crops against independent national and subnational data from 36 National Statistical Offices (NSOs), national-level crop area data for more than 180 countries and territories from FAOSTAT, as well as geospatially, against a newly available high-resolution (30 m) cropland agreement map (Tubiello et al., 2023). Results indicated robustness against the available independent information, with CROPGRIDS world total harvested and crop areas around 1.5 billion hectares. To the best of our knowledge, CROPGRIDS represents the most comprehensive update of previous work on the subject area, offering a new benchmark of global gridded harvested and crop area data for the year circa 2020. CROPGRIDS dataset can be downloaded at https://doi.org/10.6084/m9.figshare.22491997 (Tang et al., 2023).

## 1 Introduction

Detailed global geospatial information on the distribution of crop types over time is required to understand planetary boundaries and support decision-making at all scales, from land use and land use change dynamics to the impacts of agricultural



inputs use on the environment. Geo-referenced crop information is particularly valuable for improving reporting and monitoring progress at sub-national scales under the Sustainable Development Goals (SDG), in particular Goal 2 on food security, productivity and sustainability of agriculture (Tubiello et al., 2021).

The most comprehensive geospatial product available today, covering 175 crops at a resolution of 10 km globally (Monfreda et al., 2008)—henceforth referred to herein as MFR, from the initials of the authors—has nonetheless become rather outdated, providing information limited to the year 2000, whereas significant changes in cropland extent have been documented over the past twenty years (Potapov et al., 2022; Tubiello et al., 2023). MRF was created by spatially disaggregating official national and sub-national harvested area information over a gridded cropland map derived from remote

sensing. It has since been used in dozens of published studies, most notably for assessing planetary boundaries with respect to food and agriculture (Foley et al., 2011). Several crop type mapping efforts were made since the production of MRF (see Kim et al. (2021) for a comprehensive review). More recently, important initiatives such as those promoted by the European Space Agency (ESA) (Defourny et al., 2019; Franch et al., 2022) and by the USA National Aeronautics and Space Administration (NASA) were launched and are already contributing considerable progress (Lobell et al., 2018; Seifert et al., 2018; Azzari et

al., 2019). However, none of these efforts matched yet MRF in the scope and crop type coverage, so much so that many global assessments of agricultural impacts have continued to use it as a standard (Beyer et al., 2022; Ortiz-Bobea et al., 2021; Tang et al., 2021; Proctor et al., 2022).

    To update the MRF information, we produced CROPGRIDS, providing new global gridded harvested and crop area data for 173 crops circa the year 2020. CROPGRIDS was produced using MRF data as starting point, updated through

hybridisation of more recent information, i.e., merging the available, published gridded datasets for period more recent than 2000 and using a set of endogenous and exogenous data quality indicators within a multi-criteria ranking scheme to determine best-fit data by crop type and country. The resulting CROPGRIDS is a collection of harvested and crop area maps for 173 crops, at a global spatial resolution of 0.05° (approximately 5.55 km at the equator). Crop type name, harvested area and crop area definitions used in CROPGRIDS are aligned to the relevant FAO definitions (FAO, 2022). In particular, crop area refers

to FAO land use classes 'temporary' or 'permanent' crops, depending on crop type. Unlike the harvested area, multiple cropped areas of temporary crops are counted only once for crop area (FAO, 2023).

## 2 Methods

### 2.1 Input data

    We conducted a search for published peer-reviewed datasets providing geo-referenced crop-specific information,

including by grid cell: amount of harvested area ($HA$); amount of crop area ($CA$); fraction crop area ($f$, i.e., proportion of the cell occupied by crop area); or binary values ($w$) determining if a grid cell is cultivated or not. The following four minimum conditions were applied for inclusion: (1) reference year later than 2000; (2) at least one crop species also present in MRF; (3) geospatial coverage for at least one country (complete national extent); and (4) spatial resolution at least 0.083° (about 10 km



at the equator). Based on these criteria we created a library of 27 datasets, including 13 national, 8 multinational/continental,
and 6 global datasets (Table 1). Amongst the selected datasets, two provided both *HA* and *CA*, two provided only *HA*, three
provided only *f*, and 20 provided only *w* (see details in Table 1).

Additionally, we used the following datasets for data processing: a cropland agreement map (CAM) at 30 m resolution
(Tubiello et al., 2023); the FAO Global Administrative Unit Layers (GAUL) dataset (FAO, 2015); and FAOSTAT national
statistics of harvested area (FAO, 2022), see also Section 2.2.4.


**Table 1. CROPGRIDS input datasets.**

| | Acronym | Description | Reference |
|---|---|---|---|
| 1 | MRF | Global gridded *HA* [ha] for 175 crops at a resolution of 0.0833 degree (~ 10 km at the equator) in 2000. | Monfreda et al. (2008) |
| 2 | SPAM | Global gridded *HA* and *CA* [ha] for 42 crops at a resolution of 0.0833 degree (~ 10 km at the equator) in 2010. Only 30 crops considered for this work. | Yu et al. (2020) |
| 3 | GAEZ+2015 | Global gridded *HA* [ha] for 26 crops at a resolution of 0.0833 degree (~ 10 km at the equator) in 2015. Only 20 crops considered. | Grogan et al. (2022) |
| 4 | GEOGLAM | Global gridded *f* [%] for 4 crops at a resolution of 0.05 degree (~ 5.55 km at the equator) in 2020. | Becker-Reshef et al. (2022) |
| 5 | OIPA | Global gridded *w* [-] for oil palm at a resolution of 0.0000898 degree (~ 0.01 km at the equator) in 2019. | Descals et al. (2021) |
| 6 | RAP | Global gridded *w* [-] for rapeseed at a resolution of 0.0000898 degree (~ 0.01 km at the equator) in 2019. | Han, et al. (2021) |
| 7 | EU | Gridded *w* [-] for 28 countries in EU for 18 crops at a resolution of 0.0000898 degree (~ 0.01 km at the equator) in 2018. Only 12 crops considered. | d'Andrimont et al. (2021) |
| 8 | SPAMAF | Gridded *HA* and *CA* [ha] for Africa for 42 crops at a resolution of 0.0833 degree (~ 10 km at the equator) in 2017. Only 32 crops considered. | IFPRI (2020) |
| 9 | AFCAS | Gridded *f* [ha km$^{-2}$] for Africa for cassava at a resolution of 0.00833 degree (~ 1 km at the equator) in 2014. | Szyniszewska (2020) |
| 10 | SASOY | Gridded *w* [-] for South America for soybean at a resolution of 0.00025 degree (~ 0.03 km at the equator) in 2018. | Song et al. (2021) |
| 11 | MYSTHA | Gridded *w* [-] for Malaysia, Indonesia, and Thailand for oil palm at a resolution of 0.0002695 degree (~ 0.03 km at the equator) in 2017. | Danylo et al. (2021) |
| 12 | ASIARICE | Gridded *w* [-] and cropping intensity for 21 countries in Asian monsoon region for rice at a resolution of 0.0045 degree (~ 0.5 km at the equator) in 2020. | Han et al. (2022) |
| 13 | CIVGHA | Gridded *w* [-] for Cote d'Ivoire and Ghana for cocoa at a resolution of 0.0000898 degree (~ 0.01 km at the equator) in 2019. | Abu et al. (2021) |
| 14 | UZBTJK | *w* [-] for Uzbekistan and Tajikistan for 38 crops distributed as shapefile at a resolution of 0.0001 degree (~ 0.01 km at the equator) in 2015 to 2018. Only 20 crops considered. | Remelgado et al. (2020) |
| 15 | USA | Gridded *w* [-] for USA for 105 crops at a resolution of 0.0000898 degree (~ 0.01 km at the equator) in 2021. Only 64 crops considered. | Boryan et al. (2011) |
| 16 | CA | Gridded *w* [-] for Canada for 52 crops at a resolution of 0.00027 degree (~ 0.03 km at the equator) in 2021. Only 31 crops considered. | Fisette et al. (2013) |
| 17 | AFG | Gridded *w* [-] for Afghanistan for 6 crops at a resolution of 0.0000898 degree (~ 0.01 km at the equator) in 2020. Only 3 crops considered. | FAO (2021a) |
| 18 | DEU | Gridded *w* [-] for Germany for 24 crops at a resolution of 0.0000898 degree (~ 0.01 km at the equator) in 2019. Only 15 crops considered. | Blickensdörfer et al. (2022) |
| 19 | CHNWH | Gridded *w* [-] for China for winter wheat at a resolution of 0.0003 degree (~ 0.03 km at the equator) for 2018. | Dong et al. (2020) |
| 20 | CHNMZ | Gridded *w* [-] for China for maize at a resolution of 0.005 degree (~ 0.555 km at the equator) in 2017. | Qiu et al. (2018) |
| 21 | CHNMZWHRI | Gridded *w* [-] of single, double, triple cropping for China for rice, maize, and wheat at a resolution of 0.005 degree (~ 0.555 km at the equator) in 2020. | Qiu et al. (2022) |
| 22 | BGDRICE | Gridded *w* [-] of 3 growing seasons for Bangladesh for rice at a resolution of 0.0000898 degree (~ 0.01 km at the equator) in 2017. | Singha et al. (2019) |

| 23 | BRA | Gridded $w$ [-] for Brazil for sugarcane at a resolution of 0.0003 degree (~ 0.03 km at the equator) in 2019. | Zheng et al. (2022) |
|----|-----|------|------|
| 24 | SEN | Gridded $w$ [-] for Senegal for 22 crops at a resolution of 0.00009 degree (~ 0.01 km at the equator) in 2018. Only 17 crops considered. | FAO (2021b) |
| 25 | AU | Gridded $f$ [-] for Australia for 25 crops at a resolution of 0.0833 degree (~ 10 km at the equator) in 2015. Only 6 crops considered. | ABARES (2022) |
| 26 | FR | Gridded $w$ [-] for France for 11 crops at a resolution of 0.0001 degree (~ 0.01 km at the equator) in 2021. Only 5 crops considered. | Thierion et al. (2022) |
| 27 | JP | Gridded $w$ [-] for Japan for rice at a resolution of 0.0000833 degree (~ 0.01 km at the equator) in 2020. | JAXA EORC, (2021) |

## 2.2 Development of CROPGRIDS

In order to build CROPGRIDS, four steps were carried out either sequentially or in parallel (Figure 1), as follows: Step 1)

Input data harmonization; Step 2) Endogenous data quality indicators; Step 3) Exogenous data quality indicators; and Step 4) Assemblage of global maps. These steps are described in detail in the next sections. In general, the last step was achieved through a multi-criteria ranking scheme we designed using seven endogenous and two exogenous data quality indicators needed to select, for countries and territories for which data were available from multiple input datasets, the one dataset best describing a specific crop. The information collected to build CROPGRIDS spanned the period 2000-2021, with 24 out of the

27 input datasets referring to the period 2015-2021, hence, collectively referred hereafter as circa 2020.

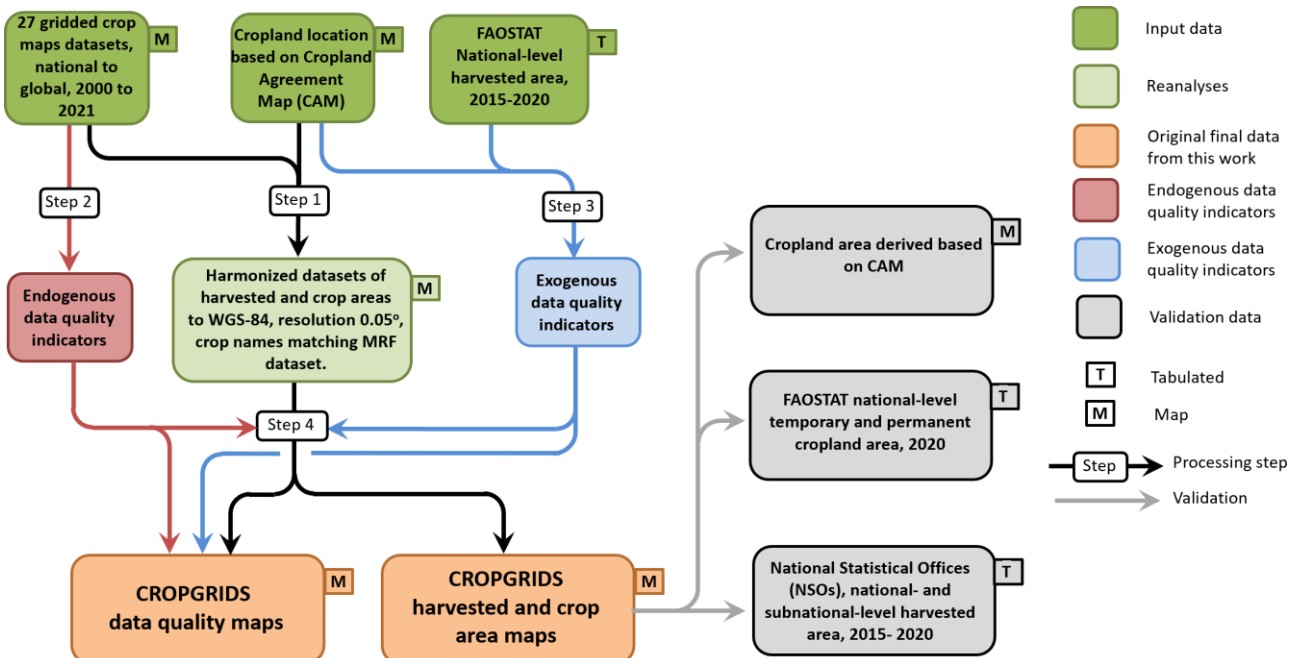

**Figure 1. Workflow of the development of CROPGRIDS.**



### 2.2.1 Step 1. Input data harmonization

The selected input datasets (Table 1) were harmonized to a common spatial resolution of 0.05° (approximately 5.55
km at the equator) using the *imresize* function in Matlab and pychnophylactic methods to ensure areal conservation
(MathWorks, 2021); and a bounding box of -180˚ to 180˚ longitude and -90˚ to 90˚ latitude using the WGS-84 coordinate
system (World Geodetic System 1984).

Prior to harmonization, we gap-filled missing *HA* and *CA* data as follows. For given *HA* data, we imputed $CA =$
$\min\{HA, GA\}$, with *GA* being the grid cell area. When only *f* was provided, we calculated crop area as $CA = f \times GA$ and
imputed $HA = CA$. When only *w* was provided, we first derived corresponding *f* values and then made the same imputations
as above. Specifically, since datasets providing *w* for individual crops had typically high spatial resolution (ranging 10 – 550
m at the equator), we performed pixel counting of *w* values to derive aggregate *f* values at the required 5.5 km resolution.
Additionally, for datasets providing *w* values over multiple growing seasons *s*, the annual *CA* was computed as $CA = \max\{CA_1,$
$CA_2, \ldots, CA_s\}$ across the season *s*; while *HA* was computed as $HA = \sum_{i=1}^{s} CA_i$. Alternatively, when the geo-referenced
cropping intensity *CI* was provided (i.e., ASIARICE, Table 1), $HA = CA \times CI$. Finally, we set a threshold for *CA* and *HA*
values, i.e., both were set to zero whenever $CA < 1$ m$^2$. Consistency diagnostics checked that $CA \le HA$, $CA \le GA$, and $CI \le 3$
(i.e., *CI* commonly less than 3, Zhang et al., 2021) were satisfied in all grid cells. In building CROPGRIDS, we harmonized
crop names in the input datasets, including performing aggregations where needed, to correspond to the crop names in MRF,
thus ensuring internal consistency and alignment with FAO crop classifications (Supplementary Table 1).

### 2.2.2 Step 2. Endogenous data quality indicators

Endogenous data quality indicators assessed both measurable and non-measurable features of a dataset that do not
depend on external information. Endogenous features considered in CROPGRIDS development included: synchrony ($Q_y$),
administration ($Q_a$), data source ($Q_s$), validation ($Q_v$), resolution ($Q_r$), maturity ($Q_m$), and type of dispatch ($Q_d$). All
endogenous features were assigned an indicator value ranging 0–1, with the end points corresponding to the lowest and highest
quality, respectively. Endogenous features were not expressed as geo-referenced maps, but rather we used them to tag
individual input datasets regardless of crop type.

$Q_y$ described the level of synchrony between the year of reference $Y_r$ of a dataset and the year of reference of
CROPGRIDS, which was set to circa 2020 (2015-2020). Specifically, datasets with $Y_r$ outside of the range 2015-2020 were
assigned a lower rank than those within the reference range:

$$Q_y = \begin{cases} \frac{Y_r - 2000}{2015 - 2000} & \text{if } Y_r < 2015 \\ 1 & \text{if } 2015 \le Y_r \le 2020 \\ 1 - \frac{Y_r - 2020}{2035 - 2020} & \text{if } Y_r > 2020. \end{cases} \qquad (1)$$



$Q_a$ described the administrative domain of a dataset (i.e., national to global). A national dataset was assigned a higher $Q_a$ value than global datasets, under our assumption that national datasets are constructed using better information from direct local knowledge. $Q_a$ was defined as:

$$Q_a = \begin{cases} 1 & \text{if national or regional} \\ 0.5 & \text{if global.} \end{cases} \qquad (2)$$

$Q_s$ described the primary data source used to develop a dataset. We assumed that datasets developed using survey data (i.e., field survey and censuses) have higher quality than those based on satellite imagery, with datasets constructed using modelling techniques having the lowest quality. We used $Q_s$ to also account for hybrid methods, assigning in such cases intermediate quality scores, as follows:

$$Q_s = \begin{cases} 1 & \text{survey, satellite, model integration} \\ 0.8 & \text{survey and satellite integration} \\ 0.7 & \text{survey and model integration} \\ 0.5 & \text{satellite and model integration} \\ 0.5 & \text{survey only} \\ 0.3 & \text{satellite only} \\ 0.2 & \text{model only} \end{cases} \qquad (3)$$

$Q_v$ was used to rank the level of validation of a dataset, against ground truth, users' feedback, statistical data, satellite images or other sources. We ranked the validation level from high to low based on the presence of field observations, the number of sources used for validation, and the separation between calibration and validation sets. $Q_v$ was defined as:

$$Q_v = \begin{cases} 1 & \text{if validated using point-scale field data with sound statistical approaches} \\ 0.5 & \text{if validated some spatial coverage with national or subnational statistics} \\ 0.25 & \text{if qualitative comparison or no attempt of validation.} \end{cases} \qquad (4)$$

$Q_r$ described the spatial resolution $r$ of a dataset. A higher rank was given to a dataset with finer resolution:

$$Q_r = 1 - \frac{r - r_{min}}{r_{max} - r_{min}}, \qquad (5)$$

where $r_{min} = 0.0000833$ degree and $r_{max} = 0.0833$ degree were the finest and coarsest resolutions across input datasets.

$Q_m$ was used to assess the level of maturity of a dataset, depending on the frequency of revisions, updates, or releases:

$$Q_m = \begin{cases} 1 & \text{if annual} \\ 0.5 & \text{if every some years} \\ 0 & \text{if never.} \end{cases} \qquad (6)$$

$Q_d$ was used to assess the level of officiality, i.e., whether a dataset was the result of an official government or non-government dispatch, assuming that official government dispatches have higher reliability than those conducted by non-government entities. It was defined as



$$Q_d = \begin{cases} 1 & \text{if government} \\ 0.5 & \text{if non-government.} \end{cases} \qquad (7)$$

All endogenous data quality indicators values are reported in Table 2 below.

**Table 2: Endogenous dataset quality indicators of all input datasets.**

| Dataset | Synchrony $Q_y$ | Administration $Q_a$ | Source $Q_s$ | Validation $Q_v$ | Resolution $Q_r$ | Maturity $Q_m$ | Dispatch $Q_d$ |
|---|---|---|---|---|---|---|---|
| MRF | 0 | 0.5 | 0.5 | 0.5 | 0 | 0 | 0.5 |
| SPAM | 0.667 | 0.5 | 0.7 | 0.5 | 0 | 0.5 | 0.5 |
| GAEZ+2015 | 1 | 0.5 | 0.5 | 0.5 | 0 | 0 | 0.5 |
| GEOGLAM | 0.867 | 0.5 | 0.5 | 0.5 | 0.400 | 0 | 0.5 |
| OIPA | 1 | 0.5 | 0.5 | 1 | 1.000 | 0 | 0.5 |
| RAP | 1 | 0.5 | 0.3 | 1 | 1.000 | 0 | 0.5 |
| EU | 1 | 1 | 0.8 | 1 | 1.000 | 0 | 0.5 |
| SPAMAF | 1 | 1 | 0.7 | 0.5 | 0.004 | 0.5 | 0.5 |
| AFCAS | 0.933 | 1 | 0.5 | 0.5 | 0.901 | 0 | 0.5 |
| SASOY | 0.933 | 1 | 0.8 | 1 | 0.998 | 0 | 0.5 |
| MYSTHA | 1 | 1 | 0.3 | 0.5 | 0.998 | 0 | 0.5 |
| ASIARICE | 1 | 1 | 0.3 | 1 | 0.947 | 0 | 0.5 |
| CIVGHA | 1 | 1 | 0.8 | 1 | 1.000 | 0 | 0.5 |
| UZBTJK | 1 | 1 | 0.5 | 0.5 | 1.000 | 0 | 0.5 |
| USA | 0.933 | 1 | 0.8 | 1 | 0.998 | 1 | 1 |
| CA | 1 | 1 | 0.8 | 1 | 0.998 | 1 | 1 |
| AFG | 1 | 1 | 0.8 | 1 | 1.000 | 0 | 0.5 |
| DEU | 1 | 1 | 1 | 1 | 1.000 | 0 | 0.5 |
| CHNWH | 1 | 1 | 0.8 | 1 | 0.997 | 0 | 0.5 |
| CHNMZ | 1 | 1 | 0.3 | 1 | 0.937 | 0 | 0.5 |
| CHNMZWHRI | 1 | 1 | 0.3 | 1 | 0.941 | 0 | 0.5 |
| BGDRICE | 1 | 1 | 0.8 | 1 | 1.000 | 0 | 0.5 |
| BRASUG | 1 | 1 | 0.8 | 1 | 0.998 | 0 | 0.5 |
| SEN | 1 | 1 | 0.8 | 1 | 1.000 | 0 | 0.5 |
| AU | 0.733 | 1 | 0.7 | 0.25 | 0.975 | 0.5 | 1 |
| FR | 0.933 | 1 | 0.3 | 0.25 | 1.000 | 1 | 0.5 |
| JP | 1 | 1 | 0.8 | 1 | 1.000 | 0.5 | 1 |

### 2.2.3 Step 3. Exogenous data quality indicators

Exogenous data quality indicators were defined to describe the quality of a dataset against independent external information. They included $Q_{CAM}$, comparison against the cropland agreement map (CAM) (Tubiello et al., 2023), and $Q_{FAO}$, comparison against FAOSTAT harvested area (FAO, 2022), average for the period 2015–2020. Unlike the endogenous indicators, exogenous data quality indicators were evaluated for each input dataset by crop and country.

     $Q_{CAM}$ was used to measure the level of agreement of the crop spatial distribution in a dataset against the FAO CAM.
We first converted the *CA* maps of each dataset and the cropland area map of CAM into binary maps, where a grid cell was assigned a value of one for non-zero crop area and zero otherwise. We then calculated $Q_{CAM}$ as:

$$Q_{CAM,i,j} = \frac{N_{overlap}(i,j)}{N_{CA}(i,j)}, \qquad (8)$$





where $N_{CA}(i,j)$ is the number of grid cells identified as crop $i$ in country $j$ in a given dataset and $N_{overlap}$ is the number of grid cells where both CAM and the given dataset have non-zero values.

150       $Q_{FAO}$ was used to measure the relative error of the input dataset crop harvested area against FAOSTAT (FAO, 2022). For crop $i$ in country $j$, $Q_{FAO,i,j}$ was defined as:

$$Q_{FAO,i,j} = 1 - \min\left\{1, \frac{|HA(i,j) - HA_{FAO}(i,j)|}{HA_{FAO}(i,j)}\right\}. \tag{9}$$

where $HA(i,j)$ is the total harvested area of crop $i$ in country $j$ in a dataset, and $HA_{FAO}$ is the corresponding 2015–2020 average
FAOSTAT value. $Q_{FAO}$ ranges between 0 and 1, with $Q_F = 1$ representing a perfect match against FAOSTAT. For countries not included in FAOSTAT, we set $Q_{FAO} = 0$.

### 2.2.4 Step 4. Assemblage of global harvested and crop area maps

Assemblage of geo-referenced harvested and crop area maps for individual crops and countries was conducted along two alternative pathways of availability: (1) only MRF data available; or (2) multiple input datasets available. In the first case,
we simply repeated the MRF information. In the latter, we used the multi-criteria ranking scheme based on endogenous and exogenous data quality indicators described above to select and use data from the dataset with the highest combined quality scores, $Q_{k,i,j}$, defined in relation to input dataset $k$ for crop $i$ in country $j$ as:

$$Q_{k,i,j} = \frac{1}{3} \times \frac{\left(Q_y + Q_a + Q_s + Q_v + Q_r + Q_m + Q_d\right)_{k,i,j}}{7} + \frac{Q_{CAM_{k,i,j}}}{3} + \frac{Q_{FAO_{k,i,j}}}{3}. \tag{10}$$

The best-fit datasets $k_{best}$ for crop $i$ in country $j$ are provided in Supplementary Table 2. For each crop, we then compiled a mosaic of *HA* and *CA* from best-fit datasets into one global map including all countries. The result of the multi-criteria analysis was that 26 out of the 27 geo-referenced datasets (excluding MYSTHA) were included in CROPGRIDS. Section 5 below provides further details on the CROPGRIDS dissemination of maps of *HA* and *CA*, as well as maps of best-fit datasets and their overall quality by crop and country.

**2.3 Quantifying uncertainty in the multi-criteria selection ranking**

The endogenous and exogenous data quality indicators used in the multi-criteria selection were assigned meaningful but arbitrary values. We conducted a Monte-Carlo analysis to quantify the uncertainty associated to such arbitrary choices, by introducing random weights, in the range 0–1, to each data quality indicator, that is: $\{w_c, w_a, w_s, w_v, w_r, w_m, w_d\}$ for endogenous and $\{w_C, w_F\}$ for exogenous indicators—whereas we implicitly had used unity weights in Eq. (10). We extracted 10,000 values
of each of the nine weights from independent Gaussian probability distribution functions with a mean equal to 1 and a standard deviation equal to 0.1 and we limited their values within the range between 0.7 and 1.3, that is three times the standard deviation. We next counted the frequency of occurrence of a selected dataset different from when using the default weight





values. This uncertainty analysis was only conducted for combinations of countries and crops where more than one dataset was available.

## 2.4 CROPGRIDS validation

We evaluated data on *HA* and *CA* in CROPGRIDS against official national and subnational statistics of crop-specific harvested area (Supplementary Table 3); FAOSTAT land areas under temporary and permanent crops by country (FAO, 2022); the pixel-level cropland areas calculated from FAO CAM (Tubiello et al., 2023).

We compiled a library of independent datasets, i.e., not used in the construction of CROPGRIDS, of national and subnational harvested area by crop from 36 National Statistical Offices (NSOs) (Supplementary Table 3), covering 69 countries and territories and 833 subnational units. Of these, national data covered 30 countries and territories and 36 reported more than 20 crops each. In total, evaluations of 121 crop data were conducted against independent crop statistics from NSOs. We matched and aggregated crop types in each NSO dataset to match those reported in CROPGRIDS. We then calculated 2015–2020 averages. We used the GAUL dataset (level 1) to identify subnational units and perform relevant aggregations from pixel level to administrative level 1. The calculations were conducted for 121 crops and were quantified using the coefficient of determination $R^2$ and normalized root mean squared errors (NRMSE) as

$$R_i^2 = 1 - \frac{\sum_{(j)}[HA(i,j) - HA_{NSO}(i,j)]^2}{\sum_{(j)}[HA(i,j) - \overline{HA(i)}]^2} \quad , \tag{11}$$

$$NRMSE_i = \frac{\sqrt{\frac{\sum_{(j)}[HA(i,j) - HA_{NSO}(i,j)]^2}{n}}}{[HA_{NSO,max}(i) - HA_{NSO,min}(i)]} , \tag{12}$$

where $HA(i,j)$ and $HA_{NSO}(i,j)$ are the harvested area of crop $i$ in administrative unit $j$ reported by in CROPGRIDS and NSOs, respectively, $\overline{HA}$ is the average of all CROPGRIDS data points, $HA_{NSO,max}$ and $HA_{NSO,min}$ are the corresponding maximum and minimum crop harvested areas of NSOs, and $n$ is the number of data points.

The crop area in CROPGRIDS was compared with FAOSTAT land areas under temporary and permanent crops for 2020 in more than 180 countries. We first classified the 173 crop types included in CROPGRIDS into temporary and permanent crops (see Supplementary Table 2 for details). We used the GAUL dataset (level 0) to identify country boundaries and perform relevant aggregations from pixel level to national level. The goodness of comparison was evaluated in terms of relative percent differences, Δ:

$$\Delta_{i,j} = \frac{CA(i,j) - CA_{FAO}(i,j)}{CA_{FAO}(i,j)} \times 100 \tag{14}$$

where $CA(i,j)$ and $CA_{FAO}(i,j)$ are the crop area in country $j$ in CROPGRIDS and FAOSTAT, respectively, with $i$ being either temporary or permanent crops.



We next validated the *CA* of all crops included in CROPGRIDS geo-spatially against the cropland area in CAM. Firstly, we calculated the sum of the *CA* of all crops in each grid cell *g* in CROPGRIDS, $CA_{total,CROPGRIDS}(g)$. Next, for each grid cell, we calculated the percent difference as:

$$\Delta CA(g) = \frac{CA_{total,CROPGRIDS}(g) - CA_{CAM}(g)}{GA(g)} \times 100,$$  (15)

where $CA_{CAM}$ is cropland area in CAM (Tubiello et al., 2023).

## 3. Results

### 3.1 Key features of CROPGRIDS

CROPGRIDS provides updates on the spatial distribution of 80 crops out of the 173 crops in MRF across more than 180 countries for the year 2020. In total, CROPGRIDS updated about 1.2 billion hectares of crop area, corresponding to 81%
of the total crop areas included in CROPGIRDS. The updates included 32 crops with more than 50% updated grid cells (Table 3 and Supplementary Table 2). Major crops featured greater than 90% updated grid cells compared to MRF, with soybean, rapeseed, and oil palm having the greatest percent update (Table 3). A total of 13 among the world top 15 crops in terms of harvested area were updated in more than 100 countries.

The information updated in CROPGRIDS indicates that in 2020 the world total harvested area (i.e., the sum across
all 173 crops) was 1.54 billion ha, while world total crop area was 1.48 billion ha. These figures overestimated the corresponding FAO data for 2020, respectively by 7% for harvested area (FAOSTAT: 1.44 billion ha) and by 20% for crop area (FAOSTAT: 1.23 billion ha, i.e., the sum of land area under temporary and permanent crops). Specifically, the global crop-specific harvested area in CROPGRIDS matched well with those reported in FAOSTAT with an $R^2 = 0.99$ (Figure 2, red markers), with 48 crops having a difference less than ±10% (Supplementary Figure 2). Comparison of CROPGRIDS against
national-level crop-specific harvested areas of FAOSTAT also shows good matching with an $R^2 = 0.97$ (Figure 2, grey markers). CROPGRIDS generally overestimated *HA* for crops and countries that have a harvested area less than 100 ha in FAOSTAT. About 40% of data points (out of a total of 8,678 pairs) had differences less than ±30%, while only less than 11% had a difference greater than ±100%.

We next estimated 2000–2020 change in world harvested area by crop by comparing MRF to CROPGRIDS data and
compared results with FAOSTAT. Results were in good agreement for 13 major crops, except for sorghum (Table 3). For instance, we estimated an increase of 23 million ha for oil palm since 2000, a 146% increase since 2000, which compared well with the 175% increase estimated with FAOSTAT (Table 3). Likewise, we estimated a 69% increase in harvested area of cassava vs. 62% by FAOSTAT, or +7% for rice vs. 6% in FAOSTAT. Conversely, we estimated a 22% increase in the harvested area of sorghum while FAOSTAT indicates a 4% decrease.

We presented examples of harvested area maps for the top four crops experiencing major changes since 2000, i.e., oil palm, soybean, cassava, and maize (Figure 3), and the corresponding best-fit datasets (Figure 4).

**Table 3. Percent of grid cells and countries being updated with spatial data more recent than 2000 for the top 15 crops with the largest global harvested area.** % grid cells updated is defined as the number of grid cells identified as a specific crop (i.e., *HA* in CROPGRIDS > 0) with data sourced from datasets more recent than 2000 over the total number of grid cells identified as that specific crop. % countries updated refers to the number of countries being updated with spatial data more recent than 2000 over the total number of countries producing that specific crop. $\Delta HA_{Grid}$ refers to the change in the world's harvested area between 2000 and 2020 estimated based on georeferenced maps in MRF and CROPGRIDS. $\Delta HA_{FAO}$ refers to the change in the world's harvested area between 2000 and 2020 estimated using FAOSTAT database.

| Crops | % grid cells updated | Number of countries updated | % of all producing countries | $\Delta HA_{Grid}$ (%) | $\Delta HA_{FAO}$ (%) |
|---|---|---|---|---|---|
| barley | 98 | 122 | 90 | -11 | -3 |
| bean | 95 | 153 | 93 | 25 | 44 |
| cassava | 96 | 111 | 90 | 69 | 62 |
| cotton | 94 | 116 | 89 | 14 | 2 |
| groundnut | 96 | 132 | 91 | 25 | 31 |
| maize | 98 | 184 | 97 | 50 | 46 |
| millet | 92 | 110 | 85 | -6 | -16 |
| oil palm | 98 | 66 | 80 | 146 | 175 |
| rapeseed | 99 | 87 | 85 | 43 | 32 |
| rice | 98 | 155 | 96 | 7 | 6 |
| sorghum | 98 | 137 | 86 | 22 | -4 |
| soybean | 100 | 142 | 95 | 75 | 71 |
| sugarcane | 93 | 114 | 86 | 35 | 37 |
| sunflower | 91 | 103 | 82 | 25 | 31 |
| wheat | 96 | 137 | 90 | 6 | 1 |


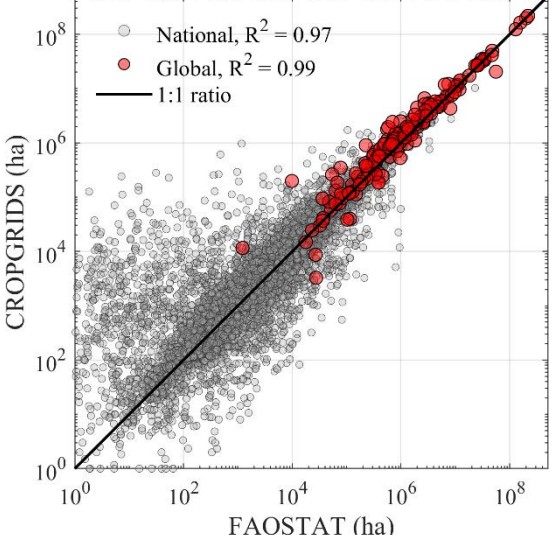

**Figure 2. Comparison of crop harvested area in CROPGRIDS against FAOSTAT.** In total, there were 8,678 pairs of comparisons at national-level and 153 pairs for global crop-specific harvested areas across.





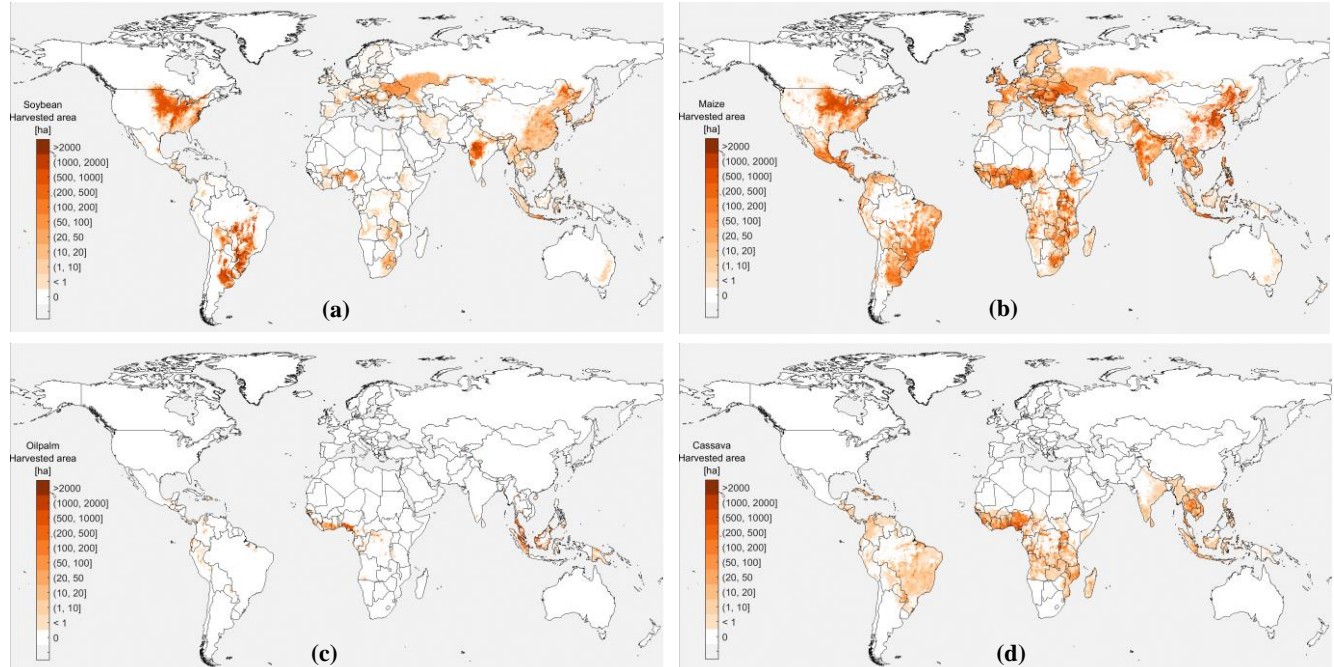

**Figure 3. Harvested area maps in CROPGRIDS for the top four crops experiencing the largest expansion since 2000.** (a) Soybean, (b) maize, (c) oil palm, and (d) cassava.





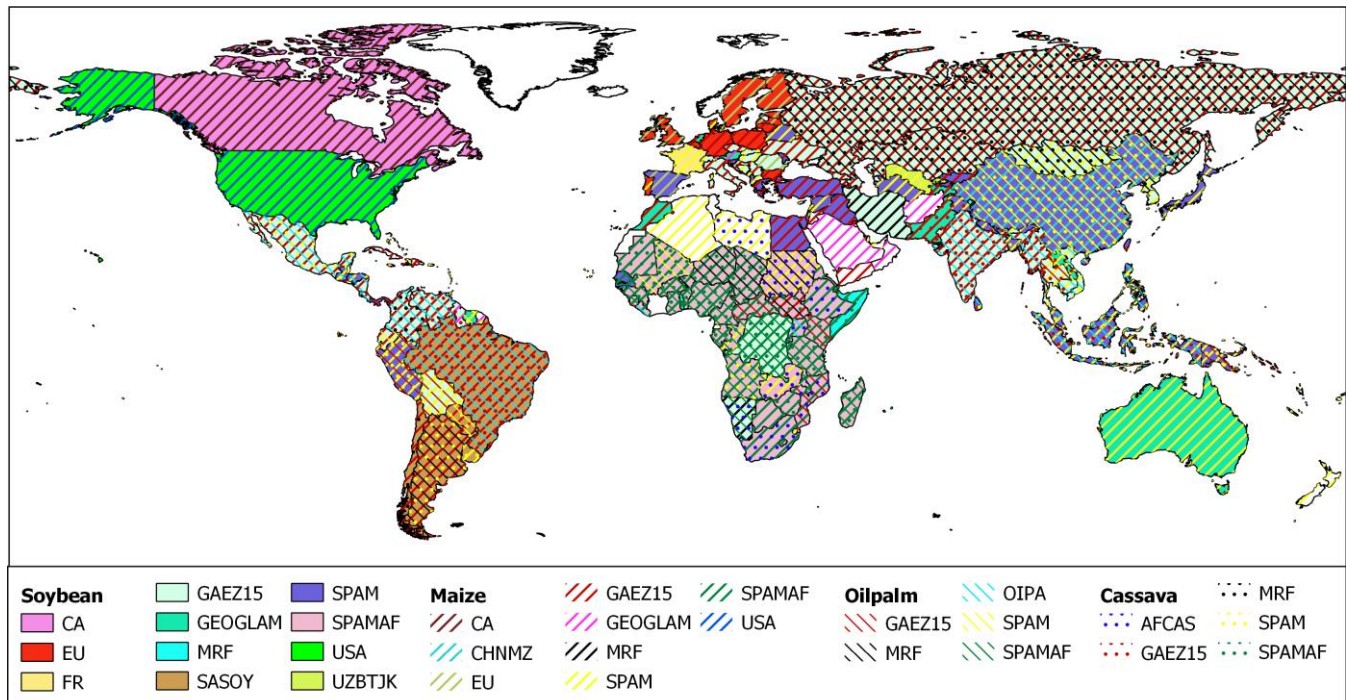

**Figure 4. Datasets used to assemble harvested and crop area maps in CROPGRIDS for soybean, maize, oil palm and cassava**.

## 3.2 Validation of CROPGRIDS against FAOSTAT land area under temporary and permanent crops

250   In CROPGRIDS, temporary crops covered 1.34 billion ha of global cropland area, overestimated by approximately 28% the temporary crop area reported in FAOSTAT for 2020, which is 1.06 billion ha. The comparison of temporary crop areas at national-level showed a relatively good match to FAOSTAT data, with differences between ±40% in 114 countries out of 188 countries compared (Supplementary Figure 1a). CROPGRIDS indicated about 45% more temporary crop area than FAOSTAT in South America (mainly Brazil and Peru) and Asia (mainly China) (Table 4, Supplementary Figure 1a).

255   Additionally, the 2020 world total permanent crop area in CROPGRIDS was 140 million ha, consistent with but lower than the 170 million ha estimated by FAOSTAT. Underestimation by CROPGRIDS of permanent crops area compared to FAOSTAT was evident in Asia (mainly Mongolia, Afghanistan, Vietnam, Cambodia), South and Central America (mainly Peru, Guatemala and Nicaragua) and the Caribbean (Table 4, Supplementary Figure 1b).




**Table 4: The comparison of land area used for cultivating temporary and permanent crops in different regions in 2020, using data**
**from CROPGRIDS and FAOSTAT.** The classification of temporary and permanent crops was reported in Supplementary Table 2.

| Region | Crop area (million hectares) | | | |
| --- | --- | --- | --- | --- |
| | Temporary crops | | Permanent crops | |
| | CROPGRIDS | FAOSTAT | CROPGRIDS | FAOSTAT |
| Africa | 203 | 193 | 31.5 | 36.3 |
| Northern America | 176 | 178 | 2.8 | 2.9 |
| Central America | 18.9 | 17.3 | 4.1 | 5.5 |
| Caribbean | 3.2 | 3.1 | 1.1 | 1.5 |
| South America | 137 | 86.9 | 12.2 | 17.9 |
| Asia | 535 | 363 | 70.7 | 93.4 |
| Europe | 238 | 194 | 14.9 | 15.6 |
| Oceania | 26.4 | 26.5 | 1.3 | 1.5 |
| World | 1337 | 1061 | 139 | 175 |

### 3.3 Comparison of CROPGRIDS crop area with cropland map

All grid cells identified as cropland (i.e., total *CA* across all crops in a grid cell > 0) in CROPGRIDS were also
identified as cropland in CAM. Globally, CAM estimated a cropland area of about 1.56 billion ha fully consistent with the
1.48 billion ha of crop area estimated in CROPGRIDS. Among all grid cells identified as cropland in CROPGRIDS, the
majority (about 71%) had a discrepancy between ± 10% of the grid cell area (Figure 5). However, in approximately 2% of the
grid cells, CROPGRIDS found a much larger cropland extent than CAM, particularly in Africa (e.g., Nigeria and Guinea) and
Asia (e.g., India and China).

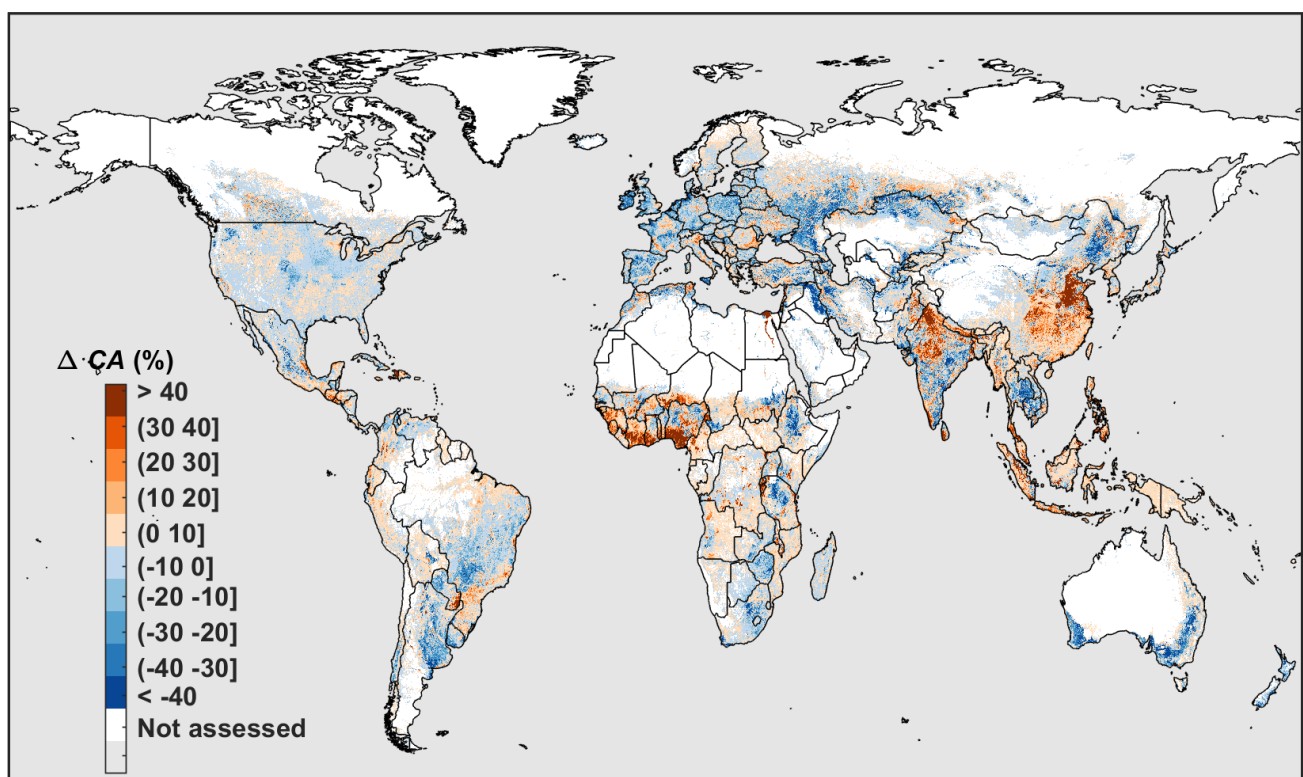

**Figure 5. Validation of the total crop area of all crops included in CROPGRIDS against the cropland area in CAM by Tubiello et al. (2023).** All grid cells in CROPGRIDS that have a crop area greater than zero were identified as cropland in CAM. Δ *CA* was calculated as the difference of crop area as a percent of grid area (see Eq. 15).


## 3.4 Validation of CROPGRIDS with official national and subnational data

We assessed CROPGRIDS against independent data sourced from National Statistical Offices (NSOs). Among the 121 crops suitable for comparison, the harvested area of 71 crops in CROPGRIDS agreed relatively well with data from NSOs ($R^2 > 0.5$, NRMSE < 0.1, Figure 6 and Supplementary Figure 3). Specifically, the comparisons for important crops such as

wheat, maize, rice, soybean, barley, rapeseed, cassava, sunflower, sugarcane, and oil palm had $R^2 > 0.91$ and NRMSE ≤ 0.05, showing very good agreement with officially reported national and subnational statistics (Figure 6). We found that for the major crops, the largest discrepancies between CROPGRIDS and national data were largely due to the GAEZ+15 dataset.

**Figure 6. Validation of crop harvested areas in CROPGRIDS against data from National Statistical Offices at national and subnational levels.** The % updated shows the percentage of grid cells for a specific crop in CROPGRIDS being updated with spatial data more recent than 2000. The colours of the markers refer to the georeferenced datasets selected to use in CROPGRIDS. This figure shows only the validation for the top 15 crops with the largest global harvested area. The validations for the other 115 crops are shown in Supplementary Figure 3.



## 3.5 Uncertainty of dataset selection

Results of the Monte-Carlo analysis on the endogenous and exogenous characteristics of the multi-criteria selection ranking scheme suggested that the method of best-fit dataset selection was highly robust. Specifically, for the 78 crops and 187 countries with multiple datasets, the probability that the selection of the best-fit dataset would change with randomized characteristics was highly unlikely (white tiles in Supplementary Figure 4). In a minor fraction of crops and countries, the probability was greater than 0.1, with only 41 out of 3346 assessed pairs of crops and countries having a probability ≥ 40%

and only 6 pairs having a probability ≥ 50% (Supplementary Figure 4).

## 4.    Discussion on limitations and uncertainties

        CROPGRIDS inherits uncertainties and errors embedded in the input datasets and these uncertainties can stem from a variety of sources. Datasets constructed based on censuses surveys (e.g., MRF and SPAM) can have uncertainties stemming from the methods used to spatialize crop area statistics at administrative-level 2 and the imperfection in statistical reporting of

crop area. Datasets constructed using remote sensing approaches can suffer from the inherent uncertainties in remote sensing data, such as, atmospheric interference and limitations in spatial resolution. More generally, these datasets also carry forward uncertainties underlying in the cropland layer maps used as their input and can be limited by the availability of ground truth data in certain regions for validation purposes. These uncertainties can propagate through the mapping process and affect the accuracy of the resulting crop area estimates in CROPGRIDS.

In addition to inherited uncertainties, the construction of CROPGRIDS also suffers from known limitations. While we have accounted for cropping intensities greater than 1 for crops with multiple harvests (e.g., rice), we have neglected dual and multi-layered cropping systems when more than one crop are grown in the same cultivated area. Information about dual cropping systems across the available datasets is limited, with only the datasets for USA and Canada providing this information. Information on multi-layered cropping systems (e.g., barley below olive trees in some Mediterranean systems) is

entirely lacking. This may lead to both underestimations and overestimations in *HA* and *CA* for some countries and some crops, leaving a knowledge gap that may be filled in in future releases of CROPGRIDS.

        At a spatial resolution of 0.05˚, a grid cell has a size of approximately 5.5 km × 5.5 km, corresponding to about 3000 ha. This leads to uncertainties in the estimated harvested and crop areas for some crops typically cultivated at smaller scales except under intensively managed systems, often monocultures. It furthermore creates uncertainty at the border between two

countries and affects in particularly the calculation of the exogenous data quality indicator $Q_{FAO}$ that compares a dataset against national-level crop harvested area reported by FAOSTAT. This border effect impacts estimates mostly in small countries in two ways. The first is when a country has zero harvested and crop areas for a crop across all datasets because border grid cells fall in the neighbouring country, whereas FAOSTAT reports non-zero values. In this case, no selection is performed. The second is when, in contrast, a country has a harvested area greater than zero when grid cells of other neighbour countries fall

within a country and FAOSTAT returns zero value. In this case, datasets will still be ranked and the best-fit will be selected

according to other quality indicators. This known bias is difficult to detect and correct, especially for small countries, because whether a border grid cell belongs to one or another country cannot be estimated correctly at the given resolution. Specifically, this bias is scale-dependent and its occurrence decreases with increasing resolution and data quality, including of the layer of administrative boundaries used to extract country statistics.

Due to constraint in spatial resolution, CROPGRIDS excludes a few small countries or territories (i.e., Falkland, Faroe Islands, French S.A.T., Heart Island, Isle of Man, Kingman Reef, Kiribati, Ma'tan al-Sarra, Mayotte, Nether. Antilles, Palau, Réunion, Saint Pierre, South Georgia, Svalbard, Virgin Islands). Greenland is also excluded, considering the small area of cultivated land. The harvested and crop areas of these countries were marked as "No data" in CROPGRIDS.

## 5.   Data availability

The CROPGRIDS data are available via *figshare* at https://doi.org/10.6084/m9.figshare.22491997 (Tang, et al., 2023). The dataset includes georeferenced cultivated area, harvested area and data quality maps for 173 individual crops, and tables reporting the crop harvested and cultivated areas in each country.

## 6.   Conclusions.

CROPGRIDS is a dataset of globally georeferenced harvested and crop area maps for 173 crops circa 2020. By 330    assimilating and harmonizing multiple peer-reviewed datasets, CROPGRIDS provides a significant update to the maps of Monfreda et al. (2008), which represented until now the most comprehensive dataset of crop area information circa the year 2000. CROPGRIDS was evaluated using multiple data sources, including a newly developed cropland agreement map (Tubiello et al., 2023), national data for temporary and permanent crop areas in more than 180 countries and territories as reported by FAOSTAT, and both national and subnational data for 121 crops from 36 National Statistical Offices. The 335    CROPGRIDS dataset will facilitate global-scale assessments in various disciplines, including agriculture and resource management, food systems, environmental impact and sustainability analyses, agroeconomics, inter-region trading, and international policy and strategies establishment.

## 7.   Author contributions.

FM, FNT, and GC designed and conceptualized the study; THN collected the data; LC provided input data for cropland 340    agreement map; FM developed the workflow and conducted the construction of the dataset; THN, FHMT, and FM analysed the data; FHMT drafted the manuscript; All authors contributed to the interpretation of the results, provided in-depth advice and commented/edited the manuscript.




## 8. Competing interests.

At least one of the (co-)authors is a member of the editorial board of Earth System Science Data. The peer review process was guided by an independent editor, and the authors also have no other competing interests to declare.

### Acknowledgments

We thank Qazi M. Amir for helping with the survey of existing data. The research work leading to CROPGRIDS was supported by the United Nations, Food and Agriculture Organization under contract (UNFAO CT34329). This project was undertaken
with the assistance of resources and services from the National Computational Infrastructure (NCI), which is supported by the Australian Government via the NCMAS 2022 and Adapter 2022 schemes awarded to F. Maggi. The authors acknowledge the Sydney Informatics Hub and the use of the University of Sydney's high performance computing cluster, Artemis. We express our gratitude for the previous contributions made to crop type mapping, and in particular, we acknowledge the pioneering work of Monfreda et al. (2008), which laid the foundation for our current research.

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
