# Peer review of "CROPGRIDS: A global geo-referenced dataset of 173 crops circa 2020"

_Earth System Science Data, 2023_

## Referee Comment (RC1)

**Review**

**Earth System Science Data**

**Title: CROPGRIDS: A global geo-referenced dataset of 173 crops circa 2020**

**Manuscript ID: Essd-2023-130**

CROPGRIDS provides an updated and comprehensive dataset of harvested and crop area maps 173 crops globally circa the year 2020, at a resolution of 0.05°(~5.55 km at the equator). It is meant to be a major update of the Monfreda et al. (2008) dataset, covering 175 crops with reference year 2000 at 10 km spatial resolution. The development of CROPGRIDS involved four steps: input data harmonization, endogenous data quality indicators, exogenous data quality indicators, and the assemblage of global maps. The input datasets used in the process included various global, national, and multinational datasets that provided crop-specific information such as harvested area, crop area, fraction crop area, and binary values indicating cultivation. The dataset has undergone evaluation using various data sources and can facilitate assessments in multiple disciplines, including agriculture, resource management, food systems, environmental impact analysis, agroeconomics, and policy development.

The CROPGRIDS dataset presented in the paper is indeed a valuable resource for global-scale assessments in various disciplines. The effort to assimilate and harmonize multiple peer-reviewed datasets to create a comprehensive dataset of harvested and crop area maps for 173 crops is commendable. The dataset covers a wide range of crops and provides georeferenced information, allowing for detailed analysis and research in agriculture, resource management, food systems, and other related fields. One of the strengths of the paper is the inclusion of an evaluation process that involves multiple data sources. The use of a cropland agreement map, national data from FAOSTAT, and data from National Statistical Offices adds credibility to the dataset and enhances its reliability. The evaluation process helps validate the accuracy and quality of the CROPGRIDS dataset, making it more robust for research and analysis.

The authors also address the issue of uncertainty in the dataset, particularly with regard to the inherited uncertainties and limitations of the input datasets. They acknowledge the potential sources of uncertainties, such as spatialization methods, statistical reporting, remote sensing data limitations, and the availability of ground truth data for validation. By acknowledging these limitations, the authors demonstrate transparency and provide a realistic assessment of the dataset's reliability.

However, I do have a few major concerns. And let me list them in the following.

- 1. My first concern is that the authors mixed all the crop type maps from different years between 2000 and 2020. The underlying assumption is that crop distribution does not change or change little over this long period and so you could use the crop in a certain location/pixel in previous year to represent the same crop in the latest year circa 2020. This is in fact not true. While cropland may not change from year to year, crop types vary from year to year due to market conditions, climate forecasting (e.g. farmers decide what to plant in their fields by looking into the expected prices and climate forecasting), and even crop rotations or fallow. 20 years is a long time. This is exactly why crop type mapping is much harder than cropland extent mapping.
- 2. Related to the above mixture is that they mix and match the crops together. Admittedly, the authors try to harmonize "crop names in the input datasets, including performing aggregations

where needed, to correspond to the crop names in MRF, thus ensuring internal consistency and alignment with FAO crop classifications (Supplementary Table 1)" (page 5 Line 97-100). In fact, these different crop maps are created for different purposes at different times, and their definition of crops, particularly aggregated ones such as beans, pulse, or even millet, may vary from one dataset to another. When the authors treat them as the same, they are mixing apples with oranges!

- 3. In crop type mapping, farming system is critical. While CROPGRIDS does address multiple cropping for certain crops, it neglects mixed or sequential (e.g. winter wheat followed by summer rice, or mixed beans and maize) cropping systems. Furthermore irrigation is another complexity. The absence of information on these systems can lead to inaccuracies in harvested and crop area estimates, potentially resulting in underestimations or overestimations. Recognizing this limitation is crucial, but it also emphasizes the need for further research and data collection to address these gaps in knowledge
- 4. For crop type mapping, the most critical input is the groundtruthing data for satellite-based mapping, sampling points for survey method, and sub-national crop statistics data for modelling-based method. CROPGIRDS doesn't add or collect any new input data (they collected independent data sourced from National Statistical Offices (NSOs) but only for validation). It assembles and integrates the existing gridded datasets. This puts extra importance or even necessity for this paper to show new innovations to be justified to be published on ESSD a data journal.
- 5. Perhaps a minor issue: circa 2020 is misleading. Normally circa 2020 implies reference year around Year 2000, having similar years of data before or after Year 2000. In fact, the majority of the global datasets in this paper are before 2018 and only a few country maps are for Years 2019 and 2020.